# Pedestrians’ Perception of Pedestrian Bridges—A Qualitative Study in Dar es Salaam

**DOI:** 10.3390/ijerph19031238

**Published:** 2022-01-22

**Authors:** Daudi Katopola, Fredirick Mashili, Marie Hasselberg

**Affiliations:** 1Department of Global Public Health, Karolinska Institutet, Tomtebodavagen 18A, 17177 Stockholm, Sweden; marie.hasselberg@ki.se; 2Department of Physiology, Muhimbili University of Health and Allied Sciences, P.O. Box 65001 Dar es Salaam, Tanzania; fredirick@gmail.com; 3National Institute of Transport, P.O. Box 705 Dar es Salaam, Tanzania

**Keywords:** pedestrian, perception, behaviors, pedestrian bridge

## Abstract

Background: About 30 percent of all road traffic deaths in Tanzania involve pedestrians. As one of the strategies to protect them, pedestrian overhead bridges have been constructed across busy roads, and plans to build more bridges are in place. It has, however, been shown that such pedestrian bridges do not necessarily discourage street-level road crossing, even when pedestrians must cross multiple lanes with heavy traffic. This paper explores the perceptions of pedestrians when crossing urban roads emphasizing pedestrian bridge users. Methods: Nineteen semi-structured interviews were conducted in situ around six pedestrian bridges in Dar es Salaam. All interviews were conducted in Swahili, recorded using digital devices, transcribed verbatim then translated into English. Content analysis was employed using qualitative data analysis software (MAXQDA). Results: We identified three overarching themes, namely, I don’t know if it’s right or wrong, they already decided; the bridge is just a crossing facility, not for other purposes; and follow your gut feeling, even if you don’t know how things will end. The results suggest that many participants prefer to look for alternative means of transport and resorted to more alternative routes just to avoid using pedestrian bridges due to bridges length and crossing time. Conclusion: These findings highlight the concerns caused by alternative uses of pedestrian bridges and underscore the importance of involving local communities and other stakeholders during planning.

## 1. Introduction

Africa including Tanzania has the highest incidence of road traffic injuries and fatalities in the world [1]. While road traffic injuries (RTIs) are decreasing in high-income countries, reports from 2016 showed an increasing trend of RTIs in most low- and middle-income countries (LMICs), a disproportionately higher rise than the number of motor vehicles available [2,3]. When compared to other road users, pedestrians are far more vulnerable to sustaining an injury when a collision occurs [4,5]. In Tanzania for instance, about 30 percent of all RTI deaths involve pedestrians [2]. Several studies show that the most common pedestrian action that results in road traffic injury is road crossing [6,7,8,9]. Likewise, a growing body of evidence shows that the majority of pedestrians prefer to cross roads using facilities on a street-level than using either pedestrian bridges or underpasses [8,10,11]. The main reasons given by pedestrians for avoiding pedestrian bridges are being in a hurry and fear of heights [12] Thus, providing a pedestrian crossing with bridges might not necessarily result in high bridge-use rate, even when pedestrians have to cross 3–4 lanes of heavy traffic [8,13,14].

In addition, studies have linked gender and age with pedestrians’ road-crossing behaviors [13], with males and older people preferring to use the street-level crossings to avoid the extra efforts required to ascend a crossing bridge [13]. Earlier studies suggest that pedestrians tend to follow their implicit attitudes, abilities, and norms embodied in regular behavior when deciding routes and the usage of pedestrian bridges [15,16]. Similarly, road crossing behavior, such as any other road safety behavior [16], is determined by the attitudes of how to use the roads, social pressure or norms, and perceived ability of the road user [17]. Often pedestrians have to sacrifice safety for convenience or vice versa when crossing the road [7,18,19]. 

There has been an increasing investment in constructing more pedestrian bridges in the city of Dar es Salaam despite the presence of limited information about their use among pedestrians. Therefore, it is unclear as to which specific attributes of behavioral intentions lead to certain road crossing behaviors among pedestrians in areas with pedestrian bridges. Moreover, it is yet unclear what perceptions, opinions, and feelings pedestrians have with regard to using pedestrian bridges especially in Sub Saharan Africa (SSA) [13].

Therefore, the main aim of this study was to gain a better understanding of the perceptions of pedestrians when crossing roads in areas with pedestrian bridges in the SSA setting. Specifically, we aim to uncover the attributes of perception among pedestrians that create road-crossing intentions resulting in various road-crossing behaviors. 

## 2. Methods

### 2.1. Study Design

This inductive semi-structured qualitative study was employed to explore the perceptions of pedestrians on pedestrian bridges in Dar es Salaam, Tanzania [20]. The study applied a phenomenographical approach in understanding pedestrians’ viewpoints on the subject [20,21]. The interviews were triangulated with multiple sources of data, such as interviews, policy documents, and the literature [22]. The study took place at all six (6) [6] pedestrian bridges that were functioning during the study period. 

### 2.2. Construction of Pedestrian Bridges

The construction of pedestrian bridges is decided upon during the pre-construction stage of the road safety audit and carried out by Tanzania National Roads Agency (TANROADS). Each audit aims at reviewing safety measures carried out at the design stage to identify potential road safety issues and opportunities for improvement used during the construction and post-construction phases [23]. The main reasons are to ease pedestrians’ crossing, improve accessibility to the BRT platform, and reduce road traffic injuries around targeted areas. 

Moreover, the archived information provided evidence of public involvement especially during the environmental and social impact assessment stage of BRT projects in the city of Dar es Salaam. Mainly a consultant is hired to conduct the mentioned awareness and consultation programs. The consultant conducts the public participation activities which involved the necessary potential interested and affected parties (I and APs) [23]. However, road users, specifically pedestrians, occasionally influence the decisions about the location and the construction of the pedestrian bridges, especially when environmental and social impact assessments are carried out [23].

### 2.3. Classification of Pedestrian Bridges

We have classified the features of the currently available pedestrian bridges in Dar es Salaam based on core similarities and differences as in Table 1: 

### 2.4. Similarities and Differences between the Pedestrian Bridges

Buguruni and Kawe pedestrian bridges are built close to densely populated locations. Both are not within the currently functioning Bus Rapid Transit (BRT) platform since the platform continues to be expanded to other suburbs of the city. Besides, the design and structure of these two bridges are similar in that they are constructed over two-way, two-lane roads with high and long barriers of metal bars in the middle to discourage pedestrians crossing on the ground level.

They also have two walking ramps (i.e., stairs and a gentle-slope lane with no stairs). The main difference between these bridges is the prevalence of economic activities within the bridge platform. It is allowed to take photos on the top of Buguruni bridge. Usually, small business activities, specifically commercial photographing, are common on the bridge. The photographers are often on shift and positioned on the bridge for seven days a week. Moreover, petty trading and resting in the shade of the bridge continue to be common. On the other hand, these socio-economic activities are not common within Kawe pedestrian bridge due to its proxy to sensitive and protected areas. 

Conversely, the rest of the bridges are wide and found within the BRT platform. They are in two-way, two-lane roads situated on two additional BRT lanes. Furthermore, Kimara and Morocco pedestrian bridges have common features such as similar design and structure, as well as being constructed at final destinations of the current BRT platform. Both have four layers of walking ramps with no stairs.

Additionally, petty trading is common on the sideways of both pedestrian bridges though photographing is not allowed unless a special request from the management of the BRT is obtained.

The other two pedestrian bridges (i.e., Manzese and Ubungo) besides being in the BRT platform, have differences in design and structure. The pedestrian bridge at Manzese is the oldest of all the bridges and has only stairs on both sides. On the other hand, the pedestrian bridge at Ubungo has stairs and long ramps on both sides and in the middle. The ramps at the middle of the bridge have been built to allow passengers to access the BRT platform. Moreover, petty trading is common in both locations while commercial photographing is more common at Manzese than at Ubungo. 

### 2.5. Ethical Considerations 

Ethical approval has been received from the Research and Publication Ethical Committee of Muhimbili University of Health and Applied Sciences (MUHAS) in Tanzania. Permission to conduct the study was sought and given by Dar es Salaam Regional office of the Tanzania National Roads Agency (TANROADS).

### 2.6. Approach 

A semi-structured interview guide was developed and used. The guide had open-ended questions (Appendix A) targeting participants’ opinions on construction, personal security, and comfort of the bridge. About 28 pedestrians were approached for interviews between January and March 2019 and asked for informed consent to participate, whereof 19 pedestrians consented to participate. Targeted participants who refused to be interviewed were either in a hurry or not feeling well enough to participate in the interviews. A convenient sampling strategy was used to recruit participants for interviews [24]. We gave a bottle of water (500 mL) to each participant for them to feel psychologically safe and relaxed during the interviews. The participants were pedestrians who either used or failed to use the bridges. These were chosen in order to maximize the likelihood of them being aware of the local conditions of pedestrian bridges, road-user behavior, road traffic regulations, and enforcement when crossing roads.

Two researchers were positioned close to the bridges and participants who consented were asked to suggest preferred and comfortable locations for interviews. The locations were either on top or alongside the bridge. This was done intentionally to give participants a sense of freedom and comfort during the interviews. 

All interviews were conducted using the Swahili language and lasted for about 40 min. The researchers had to debrief each other at the end of each day. This aimed to ensure high quality of data, the sharing of any new circumstances from the field, and the planning of next moves. We only stopped conducting the interviews when assured that saturation had been met and that no new theme was coming out from the interviews. Each interview was recorded using a digital voice recorder as well as researchers’ notepads. Thereafter, each voice clip was transcribed verbatim and documented using Microsoft Word. All transcripts were translated into English by the two researchers and later cross-checked and edited by DK to ensure reliability. 

### 2.7. Data Analysis

The analyses were conducted by the first author (DK) and the third author (MH). We used MAXQDA (Version 2018.2, VERBI Software, Berlin, Germany) software to analyze all the transcripts and photographs. All the codes were formulated by DK, checked and reviewed by MH, and both discussed the labeling of each coded event [25]. Transcripts were read and re-read to ensure effective interpretation and formation of the themes grounded in the data. We adopted in vivo coding rules to formulate all the codes. A series of discussions were involved between DK and MH to ensure clarity of all codes, thus ensuring effective descriptions of the intended message as understood from the transcripts.

## 3. Results

### Characteristics of Study Participants 

In total, 19 pedestrian participants consented and were included in the study. Table 2 shows demographic information of all participants.

We identified three overarching themes (Table 3) relating to pedestrians’ perception of pedestrian bridges and their intention to use the bridges.


**Theme 1: I don’t know if it’s right or wrong, they already decided**


This theme describes the experience of participants on how road users are involved during the planning and designing, as well as when making decisions of constructing pedestrian bridges. Questions such as what, when, where, and how were critical in describing the perceptions among the responses.


**
*1:1. We were surprised at the beginning—at the end we saw the bridge*
**


A common remark throughout the interviews was that it was a bit of a surprise to see pedestrian bridges’ construction projects through media such as newspapers, radio, and televisions. Many participants were perceived to be less directly involved in the planning, designing, and making decisions about the construction of pedestrian bridges. There were also conflicting opinions about whose responsibility it is to plan and decide. Most participants thought that it is the responsibility of engineers to decide what, where, and how the bridge should be built. 


*‘I do not think that people are involved. I think they are the decisions of the engineers. I do not know if there is anyone involved’—Female, aged 24*


Some participants were not clear about the steps that need to be taken when making decisions about building pedestrian bridges. They talked about the reasons why decision-makers fast track the process, and thus avoid involving people. They were of the opinion that maybe getting all the residents involved takes too long and perhaps the decision-makers do not want to waste that time. So, they go ahead and plan, design, and build the bridge. 


*‘I cannot know if it is right or not right because they already decided to do that, so, maybe they consider the shortage of time if they have to mobilize and consult people’—Male, aged 47*



**
*1:2. The citizens are stakeholders*
**


The participants raised many viewpoints related to the involvement of local people during the planning and design stage. For instance, one major viewpoint was about the advantages of involving local residents. In their opinion, people will have a clear and common understanding about how, when, where, and why the bridge is built. Another noted benefit was for the communities around to take care of the facilities both within and around the bridge. 


*‘[…], because the citizens are stakeholders since they are the users of the infrastructures and not the engineers. Engineers come and design the road, but they go after finishing. It is pedestrians who use it {the bridge}. Sometimes, you might face some resistance just because the final user has not been involved in the process. On the other hand, they can use it properly when involved and improperly when they are not’—Male, aged 32*


A similar concern has been noted in the national road safety policy that, in most cases, road project designs are not demand-driven, less participatory in their preparation and are unrelated to the malady they purport to cure [26].

Even though the participants felt that, although it is highly recommended to involve all groups of pedestrians in the process, they also raised that it is not always possible to get all people on board when making decisions. 


*‘It is very difficult to get all people involved talking one issue to the point of consensus’—Female, aged 24*


On the other hand, the majority of pedestrians could not propose alternative interventions to separate pedestrians from vehicles though they agree that despite the current challenges, the bridges are useful for safety. 


**
*1:3. It’s better to educate us when these structures are built*
**


The participants commented that regular awareness initiatives might be useful to encourage usage and for people to develop the right habit of using pedestrian bridges. Some participants perceived pedestrian bridges to be mainly for passengers of public transport. This type of perception could discourage the use of the bridges among pedestrians. They proposed to use media such as television and radio as well as social media in order to reach a wider audience of users.


*‘So, it would have been better if you educate us when you build these kinds of structures because some of us do not know. This would have helped’—Female, aged 42*


Furthermore, participants pointed out the role of local government to act as a platform to educate the citizens about road safety interventions. According to them, it could have added more value to involve community leadership when educating the citizens than to rely on media alone. 


*‘We can even open a class or have meetings with people and to convince the local government to discuss with the community and tell them how and when to cross the road using these pedestrian bridges. I believe it is the right way to use these local governments to convey important messages to people.’—Male, aged 28*


However, some participants perceived that it is the responsibility of everyone in the community to convey information about road safety. These participants pointed out that the use of pedestrian bridges would have increased if everyone took part in conveying road safety information. 


*‘This is a duty to all of us. We are responsible to educate and to convey messages to all the people we have targeted in the society. For example, students, small children and adults’—Male, aged 37*



**Theme 2: The bridge is just a crossing facility, not for other purposes**


This theme relates to the context in which various human activities are carried out on or around pedestrian bridges which are different from the intended purposes. One aspect of the findings regards how people use the bridges. Activities such as doing business, parking three-wheelers and motorcycles on the feet of bridges or physically exercising or resting under the shade of the bridge; standing and/or talking, as well as the prevalence of business activities such as commercial photographing on the bridge were common.


**
*2:1. Others come to spoil the meaning of the bridge*
**


Participants expressed substantial concerns about alternative uses of the bridges with regards to both planning decisions and norms. They mentioned that the emerging socio-economic activities going on around the bridges hinder the intended purpose of the bridge of facilitating pedestrians to cross roads easily with minimal risk of collision with vehicles. 


*‘This bridge has been built for pedestrians but there are others who come to spoil the meaning of this bridge, the bridge is just a passing facility and not for people who come to do their stuff, if they want to do their business they have to go down, not here’—Male, aged 32*


At the same time, some participants talked about the habit of individuals who are likely to use the shade provided by pedestrian bridges to do small businesses or to take short naps and/or rest especially during sunny hours. This finding was controversial because some participants seemed to be comfortable since the presence of socio-economic activities on or around the bridge increased their use of the bridges while others perceived that habit to be dubious especially when compared to the intended purpose of the bridge.


*‘I am here doing my small business as you can see, I cook and sell roasted maize’—Male, aged 27*


Some participants perceived pedestrian bridges as areas for doing physical exercises as noted hereunder. 


*‘For me I use this as part of my physical exercises, therefore, I run when using the stairs to make my body fit. Also, I use the other route with no stairs only when I do not feel like doing exercises’—Male, aged 28*



**
*2:2. They should not switch off lights at night*
**


Some participants had concerns about pedestrians’ safety and security due to people who just stand on the bridges especially during night hours. Many participants perceived the tendency of people to just stand around on the bridge to be frightening since some of them might be muggers. 


*‘Currently, there is enough security. Firstly, this area is within the military base thus there is no possibility for any kind of theft because normally the soldiers do their usual patrol. Therefore, security is good now, though I do not know what will happen later. But if the security lights are not switched on during the night that might result in insecurity. It is good if the lights are on for 24 h, they should not switch them off especially in the nights. This is because if someone takes you to the top of the bridge then grabs you, others might think that is romance between the partners while in fact you are being robbed by a thief’—Female, aged 35*



**
*2:3. People are crowded on the bridge*
**


Some questions in the interviews were related to whether photographing is allowed around the bridge or not, and how that influences behavioural use of pedestrians’ when accessing the bridges. Some participants perceived that the presence of commercial photographers does not in any way influence their bridge use behaviours. Additionally, others thought that it is a good thing for people to have memories through photographing in amazing places such as on or around pedestrian bridges. Thus, they talked about people’s preference to spend their free time, especially over the weekends and during public holidays, to visit places such as pedestrian bridges to take photos for the sake of memories. 


*‘People are also crowded during holidays; they take photographs for their memories’—Male, aged 37*


As explained earlier, commercial photographing was only conducted around two of the pedestrian bridges. The business was not common in the rest of the bridges unless approval has been granted by relevant authorities. All bridges along the BRT platform do not allow photographing and therefore stationed photographers are not found. 


*‘They were here in the past, but later the management of the BRT put notices prohibiting people to take photographs within the area of the platform. If someone wants to take photos, he has to get a permission from the management. Therefore, commercial photographers left since that notice started to be implemented’—Male, aged 32*



**Theme 3: Follow your gut feeling, even if you don’t know how things will end**


A recurrent theme in the interviews was a sense amongst interviewees that road crossing is not necessarily motivated by thinking since the bridges are huge structures and pedestrians could access them easily. The majority of participants were of the opinion that, it is possible for people to use pedestrian bridges subconsciously. Likewise, it was noted that participants felt that it was easy just to follow other pedestrians. 


**
*3:1. You may waste too much time if you are in a hurry*
**


It was common throughout the interviews that participants considered the length of time spent when crossing roads using pedestrian bridges. The majority perceived the distance of the bridges to be too long compared to other crossing facilities such as zebra crossing which often discourages usability by the pedestrians. Equally, others pointed out the possibility of pedestrians getting lost when using the bridges.


*‘[…], for instance, one can calculate the distance covered when using the pedestrian bridge and become discouraged to use it’—Male, aged 38*


Many other participants mentioned their feelings about their sense of comfortability and how the distance discouraged them to use the bridges. 


*‘I am not comfortable because of the walking distance. The distance is too long that you may waste too much time if you are in a hurry. For example, from the rapid bus to the other side of the road. Also, it becomes a big problem for people with health issues in their legs.’—Female, aged 24*


Similarly, something that frequently came up was that the habit to avoid using pedestrian bridges usually subjects those already vulnerable road users to a higher risk of being injured and endangers their lives as commented hereunder. Most perceived the act of going up and getting down as strenuous or tiresome.


*‘[…] they decide to cross under the bridge to avoid troubles involved when using stairs. That act of going up and down, yeah […]’—Male, aged 37*



**
*3:2. You can easily understand*
**
**, *even if you are not familiar***


Almost all participants felt that the benefits of pedestrian bridges include simplifying crossing roads, reducing road traffic crashes, and saving a life. At the same time, many of the participants were of the opinion that pedestrians do not need to use much mental effort when crossing roads as opposed to when they used to cross roads without pedestrian bridges. 


*‘[…] if you want to go to Mwenge, you walk to the top then you walk to the other end of the bridge and get down, catch a bus to Mwenge, and the same applies if someone wants to go to Tegeta. So, people find it easier and face no problems’—Female, aged 42*


Participants highlighted that the bridges are easily seen because they are huge structures, so it is easy to understand the directions. Though they pointed out that getting lost when accessing the bridge happens, still there are clear alternatives to get through. For example, through asking a fellow pedestrian or following others if one fails to ask. 


*‘[…] this bridge is easily seen because it is a huge structure. So, even if someone fails to ask how to use it, you can easily see other people using it. Where they are from and where they go. Therefore, it is a known issue unless the users are blind, then they can experience some challenges that might require to be helped. Otherwise, the structure is easily seen by us who can see. It is not hidden’—Male, aged 40*



**
*3:3. It all depends on the physical fitness of a person*
**


Most participants also discussed that the decision to use or not to use pedestrian bridges is a result of one’s perception of one’s fitness. They thought that a person who perceives him or herself to be fit and healthy would not hesitate to use the bridge as opposed to ones who perceive themselves as unfit. With that viewpoint, some participants thought that those whose health status can be considered as not fit would not have the possibility to use the bridges.


*‘When they build pedestrian bridges like this, they need to consider, […] because some other pedestrians have diabetes, blood pressure so these long distances are not friendly to them even though this is part of doing physical exercises. So, there is a need to come-up with short-distanced pedestrian bridges’—Male, aged 32*


It was also noted that it is not always about someone’s physical fitness that determines their intention of using or not using pedestrian bridges. Some other intentions are influenced by a negative attitude towards using pedestrian bridges of individuals to resist using pedestrian bridges. They thought that it is common to them to plan and opt for alternative means of transport just to avoid accessing pedestrian bridges as noted hereunder. 


*‘For example, sometimes I prefer to get off from the bus in a nearby bus-stop in order to catch another form of transport to avoid the long distances on this bridge’—Male, aged 32*


Another view that emerged throughout the interviews was that many people do usually avoid accessing the bridges including old, pregnant women, physically challenged, and sick people. For example, the physically challenged users need to find some help to push them to the top and get them down the bridge. These users have limited ability to go up and get down from the bridges using provided stairs or ramps in some of the bridges.


*‘This structure is very tiring especially to old and people who are physically challenged […]. […] even if they say that they have allowed us—the disabled to cross the road using this bridge, it is still difficult to use it. There are also old people and the sick who spend too much time getting to where they want to go. Then, they are exhausted to get to the bus station’—Male, aged 47*


## 4. Discussion

We found that the knowledge of those intending to construct pedestrian bridges do not always match pedestrians’ beliefs, intentions, and behaviours towards road crossing. As such protecting pedestrians from road crashes is more complex than protecting other road users [26]. Thus, physical protection of the pedestrians can only be applied to a very limited degree [27]. Our findings show that pedestrians understand the reasons for building pedestrian bridges and are fully aware of both individual and collective benefits. These findings confirm what has been reported in previous studies, that people know why pedestrian bridges are constructed [28,29]. The majority preferred crossing on street level even in areas with bridges, despite the knowledge of intention and benefits of pedestrian bridges. More so, unless well planned and constructed, pedestrians do not use them. The bridges are also sites of petty crime, making people prefer to cross on the road to avoid this problem. In addition, some participants reported that they looked for alternative means of transport to avoid crossing roads using pedestrian bridges especially on BRT platforms. Some people were even willing to pay extra by taking either bodaboda (i.e., commercial motorcycle) or daladala (i.e., a commuter bus) to go to either the next or previous bus station without a bridge. Avoidance of pedestrian bridges could be explained as a way to avoid the bridges when heading to pedestrians’ point of destination. This behavior could possibly be interpreted as perceived resistance which might be explained by the failure to take pedestrians’ perceptions into account when planning the bridge. 

Our results also suggest that the consequences of long pedestrian bridges may lead to ineffective use not only by old, sick, and physically challenged people [30] but also by young and energetic ones. This might imply that despite government efforts to protect pedestrians, through interventions such as constructing pedestrian bridges the earlier mentioned [31], already vulnerable road users could be more exposed to a higher risk of being injured. The assumption is that other road users mainly vehicle occupants assume that pedestrians would opt to use pedestrian bridges, so the road level is safe to drive faster than before. Similar findings have been reported in previous studies conducted in LMICs [32,33,34]. In either case, the findings postulate detailed consideration of both gender and governance dimensions versus the health of pedestrians when planning not only for pedestrian bridges but also for other built environments. 

Interestingly, as previously reported, relatively young and energetic pedestrians would avoid using the bridges [28]. In addition, other studies found that the probability of opting for a pedestrian bridge decrease as crossing time increases [12,28,35]. Further to the bridge length and crossing time, it has also been shown that people who walk in groups without talking, prefer to cross on street level rather than using the bridges [9]. Put together, these findings confirm that hazardous crossing behaviors remain a challenge not only in areas without pedestrian bridges but also in other places with crossing facilities. 

We also found that the scope of current awareness and involvement of the targeted groups before and during the construction of pedestrian bridges is perceived to be less inclusive especially to pedestrians. Generally, our study participants raised several issues and suggestions concerning current practices of public awareness and involvement initiatives. For instance, they felt that involving local government might strengthen public awareness initiatives at least at the street level. 

Of interest in the current study, we found the emergence of several alternative activities on and around the pedestrian bridges. Even though participants reported that the activities could possibly not directly influence their crossing intentions and behaviors, they contribute to more people interacting within the same space. Similarly, behaviors including those related to road crossing are determined by beliefs including behavioral, normative, and control [36]. So, the interaction of people around areas with pedestrian bridges is perceived to be a norm and it is likely that people would continue to stay and interact not only for business purposes but also for other social gatherings. However, the presence of alternative activities on or around the bridge is perceived to discourage some pedestrians from using the facilities. The main reason for discouragement could be fear of being distracted or losing personal belongings due to theft.

On the other hand, the presence of alternative human activities around the bridge could result in severe consequences if a vehicle fails and diverges from the main road for both people and their businesses.

### The Strength and Limitations of the Study

This semi-structured interview study is one of the few studies adopting a qualitative approach to uncover pedestrians’ perceptions in an LMIC setting. We used a vigorous analytical framework in order to analyze the road crossing experiences of pedestrians in a low-resource urban setting. Moreover, we applied an inductive analysis in order to expand our view from the theoretical framework with the intention of deepening understanding of the attributes [37] that affect road crossing intentions. In the same way, we believe that it is a strength that authors have various background including physiology, public health, transport, and social science.

Several limitations must be noted in this study. These include a relatively small number of participants, but all participants have experience of using pedestrian bridges and saturation was reached for all concepts [38]. We constantly validated our data using comparison analysis, meaning that returning to the data frequently in order to verify and develop the categories further. There was also the risk of respondent bias because participants were conveniently chosen and participation was voluntary, so there was a possibility for participants to be of similar characteristics. However, this was minimized by having regular debriefing discussions every day during the data collection phase to discuss challenges and come up with relevant solutions regarding the respondent’s selection. In the same vein, the input from participants can be regarded as an important contribution and as originality of the study.

## 5. Conclusions

This study implicates limited ability to use pedestrian bridges, as one of the key factors that expose already vulnerable road users to increased risks of RTI. The findings further reveal that the young, physically fit, and energetic pedestrians could encounter similar consequences. More importantly, results from this study reveal a limited involvement of pedestrians when designing road crossing facilities and underscore the importance of users’ engagement when making decisions on where and how the road crossing facilities should be constructed. In general, therefore, it is viewed that alternative uses on or around pedestrian bridges do not only discourage people but also introduce several other public health challenges due to the increased amount of waste and safety concerns. The findings reported here shed new light on the need of examining the environmental and socio-economic consequences of alternative human activities around pedestrian bridges.

## Figures and Tables

**Table 1 ijerph-19-01238-t001:** Location of pedestrian bridges in Dar es Salaam.

SN	Name of Road	Type	Characteristics	BRT Platform
1	Buguruni	2-way, 2 lanes paved trunk	30 km/h, traffic lights, zebra crossing ahead	No
2	Kawe	2-way, 2 lanes paved trunk	30 km/h	No
3	Kimara	2-way, 2 lanes paved trunk	30 km/h, zebra crossing ahead	Yes
4	Manzese	2-way, 2 lanes paved trunk	30 km/h, zebra crossing ahead	Yes
5	Morocco	2-way, 2 lanes paved trunk	30 km/h, traffic lights, zebra crossing ahead	Yes
6	Ubungo	2-way, 2 lanes paved trunk	30 km/h	Yes

**Table 2 ijerph-19-01238-t002:** Demographics of study participants.

Sex	Age	Level of Education	Years of Education	Occupation
M	38	Primary	7	Commercial motorcyclist
M	35	Ordinary secondary	11	Store keeper
M	40	Ordinary secondary	11	Bus conductor
M	27	Primary	7	Petty trader
M	39	Ordinary secondary	11	Cashier
F	33	Ordinary secondary	11	Petty trader
M	41	Ordinary secondary	11	Masonry craft man
F	22	Advanced secondary	13	Student
M	37	Diploma	14	Butcher—chicken
M	53	Primary	7	Unemployed
M	36	Ordinary secondary	11	Germ stone-mining and cutting
M	32	Ordinary secondary	11	Petty trader/motorcyclist
F	42	Primary	7	Road cleaner
M	19	Advanced secondary	13	Student
F	35	Ordinary secondary	11	Petty trader
M	28	Primary	7	Butcher
M	32	Diploma	14	Manager
F	24	Diploma	14	Student
M	47	Primary	7	Shoe shiner

**Table 3 ijerph-19-01238-t003:** Overarching themes.

I Don’t Know If It’s Right or Wrong, They Already Decided	The Bridge Is Just a Crossing Facility, Not for Other Purposes	Follow Your Gut Feeling Even If You Don’t Know What Will Happen
Sub-themes	Sub-themes	Sub-themes
We were surprised at the beginning-at the end we saw the bridge	Others come to spoil the meaning of the bridge	You may waste too much time if you are in a hurry
The citizens are stakeholders	They should not switch lights off in the night	You can easily understand, even if you are not familiar
It’s better to educate us when these structures are built	People are crowded on the bridge	It all depends on the physical fitness of a person

## Data Availability

Not applicable.

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
