# Peer review of "Pedestrians’ Perception of Pedestrian Bridges—A Qualitative Study in Dar es Salaam"

_ijerph, 2022, doi:10.3390/ijerph19031238_

Round 1

Reviewer 1 Report

The authors have addressed all my comments.

This manuscript is a resubmission of an earlier submission. The following is a list of the peer review reports and author responses from that submission.

Round 1

Reviewer 1 Report

The paper is not like a research article. It could be submitted as a technique report. Especially for the discussion and conclusions, which are not reasonable for a research paper.

Therefore, I would not able to consider it for publication in this journal. 

Reviewer 2 Report

The paper attempts to address an interesting issue that how pedestrians perceived the pedestrian bridge when crossing urban roads. The general flow is acceptable. However, I have some major concerns about the applied methods and relevant discussion.

In “Abstract”, the author claims “Content analysis was employed using a qualitative data-analysis software (MAXQDA).” I agree with that interview view method could be used for qualitative analysis. However, how could we use the results of qualitative analysis? Any new insights for policy implication? how could you generalize your findings?

In the ‘Introduction’ section, “we aim to uncover the attributes (attitudes, norms and perceived abilities) that create road-crossing intentions resulting in various road-crossing behaviours”, the attributes you used is similar to the theory of planned behavior, which you also cited numerous relevant papers. Why did you try to use the framework of TPB? I think it would provide more insightful results, from a quantitative analysis perspective.

In the ‘Results’ section, the sample size with “19 pedestrian participants” is relatively small. Like I said before, it would be hard to generalize your results with such a small sample.

I do suggest add a section “literature review”, both from methodological issue and information insights. Several recent published papers are recommended:

Vallejo-Borda, J. A., Cantillo, V., & Rodriguez-Valencia, A. (2020). A perception-based cognitive map of the pedestrian perceived quality of service on urban sidewalks. Transportation research part F: traffic psychology and behaviour, 73, 107-118.

Zhu, D., Sze, N. N., & Bai, L. (2021). Roles of personal and environmental factors in the red light running propensity of pedestrian: case study at the urban crosswalks. Transportation research part F: traffic psychology and behaviour, 76, 47-58.

Vallejo-Borda, J. A., Rosas-Satizábal, D., & Rodriguez-Valencia, A. (2020). Do attitudes and perceptions help to explain cycling infrastructure quality of service?. Transportation research part D: transport and environment, 87, 102539.

Reviewer 3 Report

The paper discusses elevated pedestrian bridges in Tanzania. As is known, these do not discourage crossing the carriageways. Given this situation, the authors conducted interviews, aimed at understanding how pedestrians are perceived and identifying possible causes of the non-use of the infrastructures dedicated to them (length of bridges, crossing time). The identified causes do not seem to me to contain other important factors, such as increased physical fatigue (climbing stairs: it is known that in the case of movement based on muscle strength, one tends to choose the route of least fatigue) and exhalations from passing cars. To encourage its use, the authors propose greater involvement during the design phase. This action can be useful for residents, but is scarcely effective for travelers. The conclusions are based on a limited number of interviews, which should be increased. The article can be published after integrating the above considerations.
